# Investigating discharge communication for chronic disease patients in three hospitals in India

Claire Humphries[1], Suganthi Jaganathan[2,3], Jeemon Panniyammakal[2,3,4], Sanjeev Singh[5], Prabhakaran Dorairaj[2,3], Malcolm Price[1,6,7], Paramjit Gill[8], Sheila Greenfield[1], Richard Lilford[9], Semira Manaseki-Holland[1]*

1 Institute of Applied Health Research, University of Birmingham, Birmingham, England, United Kingdom, 2 Public Health Foundation of India, Delhi, India, 3 Centre for Chronic Disease Control, Delhi, India, 4 Sree Chitra Tirunal Institute of Medical Sciences and Technology, Trivandrum, India, 5 Hospital Administration, Amrita Institute of Medical Sciences, Kochi, India, 6 NIHR Birmingham Biomedical Research Centre, University Hospitals Birmingham NHS Foundation Trust and University of Birmingham, Birmingham, England, United Kingdom, 7 University Hospitals Birmingham NHS Foundation Trust, Birmingham, England, United Kingdom, 8 Academic Unit of Primary Care, University of Warwick, Coventry, England, United Kingdom, 9 Centre for Applied Health Research and Delivery, University of Warwick, Coventry, England, United Kingdom

☉ These authors contributed equally to this work.
* s.manasekiholland@bham.ac.uk

**Data Availability Statement:** The minimal dataset has been uploaded to the DRYAD public repository (doi:10.5061/dryad.qnk98sfcm).

## Abstract

### Objectives

Poor discharge communication is associated with negative health outcomes in high-income countries. However, quality of discharge communication has received little attention in India and many other low and middle-income countries.

### Primary objective

To investigate verbal and documented discharge communication for chronic non-communicable disease (NCD) patients.

### Secondary objective

To explore the relationship between quality of discharge communication and health outcomes.

### Methods

### Design

Prospective study.

### Setting

Three public hospitals in Himachal Pradesh and Kerala states, India.

**Funding:** This research was funded by a joint-funded grant (Ref No: MR/M00287X/1) from the following organisations: The Department For International Development, The Economic and Social Research Council, The Medical Research Council, and the Wellcome Trust. These funders had no role in study design, data collection, analysis, decision to publish, or reporting of this manuscript. This research was also supported by the National Institute for Health Research (NIHR) Collaboration for Leadership in Applied Health Research and Care West Midlands (NIHR CLAHRC WM) and by the NIHR Birmingham Biomedical Research Centre at the University Hospitals Birmingham National Health Service (NHS) Foundation Trust and the University of Birmingham. The views expressed in this article are those of the authors and not necessarily those of the NHS, the NIHR or the Department of Health and Social Care.

**Competing interests:** The authors have declared that no competing interests exist.

## Participants

546 chronic NCD (chronic respiratory disease, cardiovascular disease or diabetes) patients. Piloted questionnaires were completed at admission, discharge and five and eighteen-week follow-up covering health status, discharge communication practices and health-seeking behaviour. Logistic regression was used to explore the relationship between quality of discharge communication and health outcomes.

## Outcome measures

### Primary

Patient recall and experiences of verbal and documented discharge communication.

### Secondary

Death, hospital readmission and self-reported deterioration of NCD/s.

## Results

All patients received discharge notes, predominantly on sheets of paper with basic pre-printed headings (71%) or no structure (19%); 31% of notes contained all the following information required for facilitating continuity of care: diagnosis, medication information, lifestyle advice, and follow-up instructions. Patient reports indicated notable variations in verbal information provided during discharge consultations; 50% received ongoing treatment/management information and 23% received lifestyle advice. Within 18 weeks of follow-up, 25 (5%) patients had died, 69 (13%) had been readmitted and 62 (11%) reported that their chronic NCD/s had deteriorated. Significant associations were found between low-quality documented discharge communication and death (AOR = 3.00; 95% CI 1.27,7.06) and low-quality verbal discharge communication and self-reported deterioration of chronic NCD/s (AOR = 0.46; 95% CI 0.25,0.83) within 18-weeks of follow-up.

## Conclusions

Sub-optimal discharge practices may be compromising continuity and safety of chronic NCD patient care. Structured protocols, documents and training are required to improve discharge communication, healthcare integration and NCD management.

## Introduction

Continuity of care can most simply be defined as: "the seamless provision of healthcare between settings and over time".[1] It is crucial for managing patients with chronic non-communicable disease (NCDs), who often require regular check-ups and care episodes across a variety of healthcare settings. Continuity of care is built around effective communication between healthcare professionals (HCPs) and between HCPs and patients. This is imperative during transitions of care, where poor handover communication can have far-reaching consequences such as delays in care, incorrect treatment and readmission.[2, 3]

Hospital discharge is one point of care transition that has proven to be particularly critical for continuity and safety of patient care. Evidence from high-income countries (HICs) has indicated that one in five patients experience an adverse event following discharge and that one-third of these events are preventable.[4] Regarding the impact of discharge communication for patient safety, there is an established link between the deficient exchange of documented information between hospital and primary care HCPs and adverse patient outcomes. [4–7] Ineffective communication between HCPs and patients/carers during discharge consultations, particularly regarding condition and/or ongoing treatment needs, has also been identified as an issue that can lead to patient misunderstanding and adverse events such as medication errors and unplanned readmissions.[4, 8–10] With regard to interventions, the literature has continually advocated the timely and accurate exchange of patient-specific information to improve coordination and safety during care transitions.[11, 12] Effective discharge education and documented summaries are particularly vital tools that have been shown to reduce a number of post-discharge complications and unplanned readmissions.[7, 13–18] However, in practice the delivery of discharge instructions often remains rushed and essential details for facilitating continuity of care such as diagnosis, medication, lifestyle and follow-up information are not always exchanged.[13, 19–24]

Despite the growing body of literature on quality of discharge communication and its impact on HCP and patient-related outcomes in HICs, similar evidence from low and middle-income countries (LMICs) remains relatively scarce.[25] Some single-site observational studies have evaluated discharge practices and found issues regarding deficient documentation, guidelines, standardised procedures and patient education.[26–30] In addition, a recent (2019) study from South Africa found inadequate discharge planning (a process which involves healthcare information transfer between HCPs and between HCPs and patients/carers to ensure coordination and continuity of care) to be a significant contributor to potentially avoidable causes of readmissions.[31] Across India, a handful of studies have found inconsistencies in the provision of discharge information via HCP and patient reports and discharge ticket evaluations.[32–34] In addition, a study in an Indian hospital emergency department reported improvements in recorded discharge information following the implementation of pre-formatted discharge summaries.[35]

The importance of investigating factors affecting continuity of care in LMICs and India, in particular, is increasing due to the rising prevalence of chronic NCDs, which require sustained care across settings.[36] Given the resource constraints across numerous LMIC settings, the need to elucidate context-relevant strategies to improve patient self-management and avoid unnecessary healthcare utilisation is also vital. In the study areas of India (i.e. Himachal Pradesh and Kerala states), a lack of standardised information systems within and between levels of healthcare has resulted in patient-held medical information serving as the predominant vehicle for handover communication between HCPs and between HCPs and patients.[37, 38] As well as verbal advice/instructions, patients are often provided with medical documents (including discharge notes) during healthcare visits that can be transported between HCPs whilst seeking care from a variety of public and private providers. Therefore, the quality of both verbal and documented information exchanged between HCPs and patients is critical in facilitating adequate comprehension, coordination and continuity of care.

We conducted a prospective cohort study with the primary objective of investigating verbal and documented discharge communication for chronic NCD patients in three hospitals in Himachal Pradesh and Kerala states, India. In addition, given the increasing evidence on the significance of information exchange for patient outcomes, a secondary objective was to explore the relationship between quality of discharge communication and adverse health outcomes.

## Methods

### Study setting

**Overview.**  The study was conducted from December 2014 to November 2015 in Himachal Pradesh and Kerala states. Patients were recruited and initial data was collected from three hospitals: one rural secondary-care hospital (150 beds) in the district of Solan, Himachal Pradesh, and one peri-urban secondary-care hospital (150 beds) and one urban tertiary-care hospital (783 beds) in the district of Ernakulam, Kerala. These settings were selected to capture a range of contrasting hospital types and environments. We selected public rather than private healthcare facilities for this study as this is where targeted improvements to health systems could be implemented more systematically (i.e. via state health departments). In addition, public facilities are where a large proportion of socio-economically vulnerable populations access healthcare in India.

**Public healthcare in India.**  At the national level, public healthcare in India is directed by the Ministry of Health and Family Welfare. At the state level, public healthcare is managed by the State Department of Health and Family Welfare, which has considerable autonomy in deciding upon, designing and delivering health programs. Fig 1 contains a summary of the structure of public healthcare in India, based on Indian Public Health Standard Norms.[39]

**Population and public healthcare in Solan, Himachal Pradesh and Ernakulam, Kerala.**  Himachal Pradesh is a predominantly rural and mountainous state in northern India with a population of 6.86 million people.[40] Solan district has a population of approximately 580,000 people, with 82.4% living in rural areas. There average literacy rate across the district is 83.7%, which is higher than the national average (74%) but rates remain lower for women compared to men (77.0% vs. 89.6%, respectively).[41] A recent study regarding the availability of health services across Himachal Pradesh found that there are 5 hospitals, 6 Community Health Centres (CHCs), 33 Primary Health Centres (PHCs) and 179 Sub Health Centres (SHCs) in Solan District. With regard to primary care infrastructure, it was calculated that there are approximately 0.83 CHCs per 80,000 persons, 1.14 PHCs per 20,000 persons and 0.93 SHCs per 300 persons.[42]

Kerala is a state in the south-west of India with a population of 34.8 million people and a greater than national average urban-based population of 47.7%.[43] Ernakulam district has a population of approximately 3.2 million people, with 68.1% living in urban areas. The average literacy rate across the district is 95.89%, which is notably higher than the national average with similar rates between women and men (94.5% vs. 97.4% respectively).[44] Regarding the availability of health services, the Kerala Department of Health Services (DHS) reports there are 15 hospitals, 23 CHCs, 75 PHCs and 410 SHCs in Ernakulam district.[45] Based on census and DHS data, it can be calculated that there are approximately 0.58 CHCs per 80,000 persons, 0.47 PHCs per 20,000 persons and 0.04 SHCs per 300 persons in Ernakulam.

### Ethics approval

This study was reviewed and approved by the Centre for Chronic Disease Control Independent Ethics Committee, India, and the Amrita Institute of Medical Sciences Institutional Ethics Committee, India. Data archives are stored at the University of Birmingham, in accordance with the University's code of practice.

### Patient recruitment

Patients were recruited consecutively by trained social work graduate researchers (n = 6) six days per week (spread across all days of the week over the study period) between the hours of 8 am and 6 pm, as this is the window within which patients were typically discharged from study

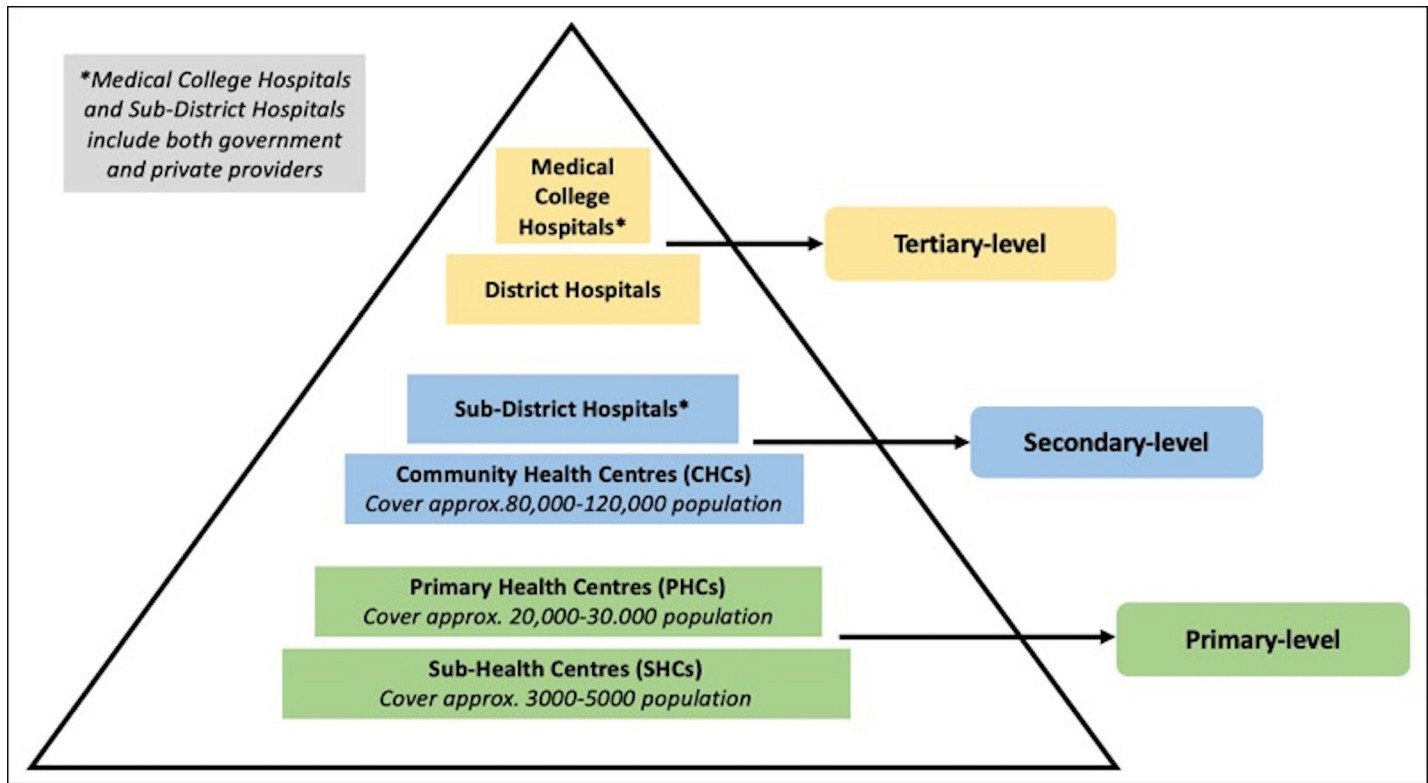

**Fig 1. Structure of the Indian public healthcare system according to Indian Public Health Standard Norms.**

hospitals. Patients were included within the first 24 hours of their admission if they met the following inclusion criteria: adults (18yrs+) with one of the following tracer chronic NCDs; cardiovascular disease, chronic respiratory disease, diabetes mellitus or hypertension. Researchers approached ward nurses to identify eligible chronic NCD patients; patients were excluded they lived outside the study areas (where planned follow-up visits would be difficult to achieve), or if ward nurses judged them to be too unwell to participate due to physical and/or cognitive impairment. Researchers then approached eligible patients (and their carers), provided them with study information sheets and verbally informed about the purpose of the research. Written consent was obtained from literate patients. For illiterate patients, oral consent was obtained along with a thumbprint and signature from a literate witness (i.e. carer) in line with World Health Organisation ethical guidelines.[46]

Due to the lack of prior work completed in the field of study, the sample size for this study was calculated using a formula to determine the minimum sample size needed to estimate a population mean with confidence limits of 5% for a variable (such as the proportion of patients receiving complete healthcare information) with a prevalence of 50%. The formula used was: Necessary Sample Size = (Z-score)$^2$ * StdDev*(1-StdDev) / (margin of error)$^2$.[47] Based on our parameters this worked out as 384.16. Therefore, we aimed to collect survey data from a minimum of 385 patients.

## Data collection

**Questionnaire.** The questionnaire used in this study was developed from one used in previous handover research conducted in Nigeria for a masters in public health thesis (see

"S1 Appendix" for a copy of the questionnaire). It was adapted using relevant handover communication literature and input from expert researchers in the UK and India. Prior to the commencement of data collection, a small pilot study was conducted in Kerala to test the questionnaire. This was conducted iteratively over three rounds, with field researchers visiting the study hospitals and interviewing two patients/carers at each location. In addition to asking questions, researchers also asked for feedback on the clarity and contextual appropriateness of the materials. Once this was completed, all researchers met with a supervisor to discuss the aim of each question, patient/carer responses and subsequently make minor amendments to the wording and structure of the questionnaire to improve clarity. After three rounds of piloting, based on both patient/carer and researcher feedback, the questionnaire was considered to be suitable for use; piloted cases were not included in the main study.

Questionnaires were completed at four time points: 1. during hospital admission; 2. immediately after hospital discharge; 3. five weeks after discharge and 4. eighteen weeks after discharge. All questionnaires were administered in person by trained social work graduate researchers (n = 6) in the form of an interview; each question was read aloud and researchers ticked the appropriate box/es for responses that corresponded to pre-defined answers and/or wrote free-text notes for responses that did not correspond to pre-defined answers. Whilst patients predominantly provided responses to questions, their accompanying carer/s (e.g. friends, family members etc.) were able to provide support for clarifying answers when required. The admission and discharge interviews were completed in the hospital, whereas the five-week and eighteen-week follow-up interviews were completed in the community.

During admission, data was collected on patient demographics, health status, previous HCP visits, referral/s (if any) and experiences of healthcare information transfer. Following hospital discharge consultations, information was collected about verbal recall of healthcare information given to patients, attitudes towards such information and subsequent plans for HCP follow-up. Researchers also assessed patient/carer understanding of their health condition and post-discharge care requirements by asking them to explain this information and then checking it against their discharge notes, or if these did not contain enough information/ were not immediately available, a ward nurse who was aware of their condition and care requirements. Researchers also evaluated the contents of all notes, prescriptions and/or other pieces of paper that the patient had brought with them to hospital and that had been given to them by a hospital HCP during admission and/or their discharge consultation. These content evaluations were completed using structured checklists within the questionnaires; space was provided for free-text entry to cover any information not included in the checklists. At five and eighteen-week follow-up visits data was collected about post-discharge adverse health outcomes, health status and health-seeking behaviour.

## Analysis

Descriptive statistics were used to outline patient demographics, attitudes and experiences of discharge communication and post-discharge health outcomes. Descriptive statistics were also used to calculate the number of patients who received all items of "key" documented and verbal information during discharge communication. For documented information, key items included: Diagnosis, medication information, lifestyle advice and follow-up instructions. For verbal information, key items included: Ongoing treatment/management information, medication information, lifestyle advice and follow-up instructions. These items were selected based on common themes across the literature regarding critical information needed to improve discharge communication and care transitions for patients with chronic conditions. [48–51]

Due to a relatively small number of outcome events in the study sample, the Firth method of multiple logistic regression was used to explore the relationship between quality of discharge communication and health outcomes at five and eighteen-weeks follow-up.[52, 53] The principle outcome variables were death (all-cause mortality), hospital readmission and self-reported deterioration of NCD/s (that the patient was hospitalised for). For explanatory variables, the completeness of key documented and verbal information was utilised to measure the quality of discharge communication. The explanatory variables of interest in this study were receiving low-quality documented discharge communication (i.e. notes containing 0–2 items of key documented information—compared to notes containing 3–4 items) and receiving low-quality verbal discharge communication (i.e. recall of receiving 0–2 items of key verbal information—compared to recall of receiving 3–4 items) during discharge consultations.

Multiple logistic regression models were employed in order to adjust for the following potential confounders: sex, age, education level, employment status, time taken to reach the hospital, number of chronic NCDs and hospital site. A count of chronic NCDs was used due to a lack of available data regarding primary diagnosis or severity of comorbidities, which would have enabled the creation of a comorbidity index. Such an approach has been validated and shown to add predictive value for survival analyses (vs. age alone).[54]

## Results

### Demographics and medical conditions

A total of 546 inpatients completed questionnaires; 305 men and 241 women. The majority of participants were aged 60 years or older (59%) and were literate with a complete primary school-level education or more (67%). Of the four chronic NCDs captured by this study, the most frequently reported was chronic respiratory disease (45%) (Table 1). See "S2 Appendix" for participant demographic information and health outcomes by study site.

### Hospital discharge

**Patient recall of verbal discharge communication.**  Most patients (89%) reported having their health condition explained to them during admission. Post-discharge care advice appeared to vary notably, as just over half (50%) of all patients recalled being given information regarding ongoing treatment/management and 23% of patients recalled receiving lifestyle advice. With regard to follow-up instructions, the majority of patients (85%) recalled being told to return for an outpatient check-up. Overall, just 15 (3%) patients recalled receiving all key verbal discharge information (i.e. ongoing treatment/management information, medication information, lifestyle advice and follow-up instructions). With regard to patient understanding of post-discharge care information, a quarter of patients/carers (25%) were judged by researchers to have a good understanding of almost all important details (Table 2).

**Patient follow-up plans.**  When asked about follow-up plans, the majority of patients (82%) stated that they planned to return to the outpatient clinic of the same hospital they were being discharged from. A notable proportion of patients also (13%) stated that they would only return to a HCP when they become unwell again (Table 2). When patients were asked about how they would explain to the next HCP what was done for them during admission, the most common response (45%) was that they had asked the doctor to explain key information to their carer/family member (Table 2).

**Documented discharge communication.**  All patients were provided with a document containing handwritten notes during their discharge consultation. Overall, most patients (94%) felt it was important to receive discharge notes, the most common reason given for this was because it helps patients to understand and explain their condition (64%). The type of

**Table 1. Participant demographic and adverse health outcomes information.**

| Characteristic | Total (n = 546) |
|---|---|
| | Frequency (%) |
| **Sex** | |
| Male | 305 (55.9) |
| Female | 241 (44.1) |
| **Age group (Years)** | |
| 18–49 | 98 (17.9) |
| 50–69 | 296 (54.2) |
| ≥70 | 152 (27.8) |
| **Level of education** | |
| Illiterate | 91 (16.7) |
| Literate with partial or completed primary school education | 258 (47.3) |
| Complete secondary school education | 132 (24.2) |
| Complete higher school/vocational studies | 52 (9.5) |
| University graduate or above | 13 (2.4) |
| **Employment status** | |
| Employed | 164 (30.0) |
| Unemployed | 369 (67.6) |
| Retired | 11 (2.0) |
| No data* | 2 (0.4) |
| **Time taken to reach hospital** | |
| <1 hour | 311 (57.0) |
| 1–4 hours | 230 (42.1) |
| >4 hours | 4 (0.7) |
| No data* | 1 (0.2) |
| **Chronic NCDs[†]** | |
| Diabetes | 157 (28.8) |
| Cardiovascular Disease | 218 (39.9) |
| Chronic Respiratory Disease | 247 (45.2) |
| Hypertension | 171 (31.1) |
| **Number of chronic NCDs (per patient)** | |
| 1 | 365 (66.9) |
| 2 | 128 (23.4) |
| 3 | 40 (7.3) |
| 4 | 13 (2.4) |
| **Adverse health outcomes at 5-week follow-up** | |
| Death (all-cause mortality) | 19 (3.5) |
| Hospital Readmission | 33 (6.0) |
| Self-reported deterioration of NCD/s | 39 (7.1) |
| No data* (loss to follow-up)[§] | 13 (2.4) |
| **Adverse health outcomes at 18-week follow-up** | |
| Death (i.e. all-cause mortality) | 25 (4.6) |
| Hospital Readmission | 69 (12.6) |
| Self-reported deterioration of NCD/s | 62 (11.4) |
| No data* (loss to follow-up)[§] | 14 (2.6) |

* No data = missing responses

†Please note that participants could select more than one answer for this question

§Patients lost to follow-up were those who could not be contacted or found during community visits in the follow-up period

Table 2. Patient recall of verbal discharge communication and follow-up plans.

| Hospital discharge | No. (n = 546) | % |
|---|---|---|
| **Health condition explained to patient during admission** | | |
| Yes | 485 | 88.8 |
| No | 38 | 7.0 |
| Patient unsure | 16 | 2.9 |
| No data* | 7 | 1.3 |
| **Post-discharge care information provided by hospital doctors[†]** | | |
| Instructions to go for further test/s | 19 | 3.5 |
| Details regarding ongoing management | 274 | 50.2 |
| Details of prescribed course of medication to be taken (and reviewed when completed) | 297 | 54.4 |
| Lifestyle advice (i.e. regarding exercise, diet, tobacco and/or alcohol) | 123 | 22.5 |
| Instructions to take rest | 1 | 0.2 |
| Instructions to visit a physiotherapist | 14 | 2.6 |
| Referral to another HCP | 3 | 0.5 |
| Patient unsure of what advice was given (if any) | 12 | 2.2 |
| No advice given | 1 | 0.2 |
| No data* | 10 | 1.8 |
| **Follow-up instructions provided by hospital doctors[†]** | | |
| Visit/s to the outpatient department of this (same) hospital | 430 | 78.8 |
| Visit/s to the outpatient department of another hospital/specific doctor | 33 | 6.0 |
| Patient unsure of what advice was given (if any) | 75 | 13.7 |
| No data* | 16 | 2.9 |
| **Patient recalled receiving all key verbal discharge information[§]** | **15** | **2.7** |
| **Patient/carer understanding of health condition and post-discharge care requirements** | | |
| Patient/carer had a good understanding of almost all important information | 135 | 24.7 |
| Patient/carer had a broadly correct understanding of important information | 186 | 34.1 |
| Patient/carer had only a basic understanding of some important information (e.g. diagnosis/medicine) | 85 | 15.6 |
| Patient/carer had very little/no understanding of any important information | 20 | 3.7 |
| No data** | 120 | 22.0 |
| **Patient plans for follow-up HCP visit/s[†]** | | |
| Return to same hospital outpatient clinic | 448 | 82.1 |
| Another government hospital | 12 | 2.2 |
| Government primary care centre | 20 | 3.7 |
| Private hospital/nursing home | 6 | 1.1 |
| Local private doctor/nurse | 6 | 1.1 |
| Physiotherapist | 1 | 0.2 |
| Traditional healer | 1 | 0.2 |
| Patient plans to only return to a HCP when they are sick again | 68 | 12.5 |
| No data* | 8 | 1.5 |
| **How patients plan to explain to next HCP what was done for them during admission[†]** | | |
| Patient asked the doctor to explain to their carer/family member | 247 | 45.2 |
| Patient asked the doctor to explain to them so they can tell the HCP when they see them | 82 | 15.0 |
| Doctor gave the patient/carer a note or discharge summary to take back to their local HCP | 116 | 21.3 |
| Patient plans to return to this (same) hospital where their medical records are stored | 100 | 18.3 |
| Patient is unsure what they will do because they cannot remember what the doctor said | 2 | 0.4 |

(*Continued*)

**Table 2.** (Continued)

| Hospital discharge | No. (n = 546) | % |
|---|---|---|
| No data* | 23 | 4.2 |

* No data = missing responses

† Participants could select more than one answer for this question

§ Ongoing management information, medication information, lifestyle advice and follow-up information

**No data = missing responses; please note the larger number of missing responses to this question was due to a lack of available documented discharge information and/or ward nurses, which were required at the time of questioning for the researcher to verify patient/carer understanding of their condition and post-discharge care requirements

discharge documents provided to patients varied; the majority received either a sheet of paper with basic pre-printed headings (71%—see "S3 Appendix" for a photographed example) or an unstructured (i.e. otherwise blank) sheet of paper (19%). The contents of discharge notes received by patients varied greatly between individuals, with only 31% containing all four items of key healthcare information required for effectively facilitating transitions of care (i.e. diagnosis, medication information, lifestyle advice, and follow-up instructions) (Table 3).

## Five-week follow-up

Five weeks post-discharge 13 (2%) patients had been lost to follow-up, 19 (4%) patients had died, 33 (6%) patients reported they had been re-admitted to hospital and 39 (7%) patients reported a deterioration in their health (related to the NCD/s they were hospitalised for) (Table 1). See "S4 Appendix" for flowchart summarizing participant inclusion and exclusion throughout the study.

Results of the adjusted analyses of the association between low-quality discharge communication and death, hospital readmission and self-reported deterioration of NCD/s within 5 weeks of discharge are presented in Table 4 (see "S5 Appendix" for unadjusted results). They showed significant ($p<0.05$) increased odds of death within five weeks of discharge for patients who received low-quality discharge notes (AOR = 4.43; 95% CI 1.46, 13.46). No other significant associations were found. Goodness-of-fit tests using the method of Heinze and Schemper were performed for each adjusted multivariate analysis and are reported in "S6 Appendix". In this approach, coefficients are constrained to zero and left in the model in order to allow their contribution to the penalization.[55]

## Eighteen-week follow-up

Eighteen weeks post-discharge 14 (2%) patients had been lost to follow-up, 25 (5%) patient had died, 69 (13%) patients reported that they had been readmitted to hospital and 62 (11%) patients reported a deterioration in their health (related to the NCD/s they were hospitalised for) (Table 1).

Results of the adjusted analyses of the association between low-quality discharge communication and death, hospital readmission and self-reported deterioration of NCD/s within eighteen weeks of discharge are presented in Table 4 (see "S6 Appendix" for unadjusted results). They showed significant ($p<0.05$) increased odds of death for patients who received low-quality discharge notes (AOR = 3.00; 95% CI 1.27, 7.09). With regard to verbal information, the results showed a significant ($p<0.05$) decreased odds of self-reported deterioration of NCD/s within eighteen weeks of discharge for patients who recalled receiving low-quality verbal discharge communication (AOR = 0.48; 95%CI 0.27–0.87). No other significant associations were found (see "S6 Appendix" for goodness-of-fit statistics for all adjusted analyses).

**Table 3. Documented discharge communication.**

| Hospital discharge | No. (n = 546) | % |
|---|---|---|
| **Documented information given to patients at discharge** | | |
| Patient received discharge document/s (seen by a researcher) | 546 | 100 |
| **Patient attitudes regarding importance of receiving discharge document/s** | | |
| It is important to receive discharge document/s | 513 | 94.0 |
| It is not important to receive discharge document/s | 24 | 4.4 |
| Unsure whether it is important or not to receive discharge document/s | 6 | 1.1 |
| No data* | 3 | 0.5 |
| **Reasons given for why patients feel it is important[†]** | **No. (n = 513)** | **%** |
| It helps to understand and explain my condition/s | 330 | 64.3 |
| It helps me to get attended to faster at my next HCP visit | 85 | 16.6 |
| I feel it's more professional | 19 | 3.7 |
| I have to submit this for claiming insurance | 89 | 17.3 |
| It will help in an emergency | 2 | 0.4 |
| It is a helpful medical identification certificate | 1 | 0.2 |
| **Reasons given for why patients feel it is not important[†]** | **No. (n = 24)** | **%** |
| The notes get lost | 5 | 20.8 |
| Everyone receives the same standard of care regardless | 19 | 79.2 |
| **Types of documents given to patients at discharge** | | |
| Discharge booklet | 42 | 7.7 |
| Structured discharge document (i.e. form/card with basic pre-printed headings) | 386 | 70.7 |
| Unstructured discharge document (i.e. note/letter on otherwise blank sheet of paper) | 104 | 19.0 |
| Prescription card (containing medication information only) | 5 | 0.9 |
| Referral letter | 1 | 0.2 |
| No data* | 8 | 1.5 |
| **Contents of discharge documents** | **No. (n = 545)[§]** | **%** |
| Illegible notes | 29 | 5.3 |
| Name of doctor/contact at the hospital | 379 | 69.5 |
| Date | 517 | 94.9 |
| Name, age and sex of patient | 523 | 96.0 |
| Diagnosis | 536 | 98.4 |
| Medication information | 477 | 87.5 |
| Follow-up instructions | 299 | 54.9 |
| Lifestyle advice (e.g. exercise, diet, tobacco, alcohol etc.) | 268 | 49.2 |
| Past medical history for current condition | 331 | 60.7 |
| Past medical history for other conditions | 99 | 18.2 |
| Patient's signs, symptoms and problems when admitted | 506 | 92.8 |
| Tests performed during admission (without results) | 98 | 18.0 |
| Tests performed during admission (with results) | 429 | 78.7 |
| **Discharge document contained all key items of information[**]** | **168** | **30.8** |

* No data = missing responses

† Participants could select more than one answer for this question

§ One patient did not give permission for the contents of their discharge document/s to be analysed

** Diagnosis, medication information, lifestyle advice and follow-up instructions

**Table 4. Adjusted associations between receiving low-quality discharge communication and the likelihood of experiencing adverse health outcomes within five and eighteen weeks of discharge.**

| Death within 5 weeks of discharge | Adjusted odds ratios* | | | Death within 18 weeks of discharge | Adjusted odds ratios* | | |
|---|---|---|---|---|---|---|---|
| | OR | 95% CI | p-value | | OR | 95% CI | p-value |
| *No. of items of key documented discharge information†* | | | | *No. of items of key documented discharge information†* | | | |
| 0 to 2 items | 4.37 | 1.46–13.11 | 0.009§ | 0 to 2 items | 3.00 | 1.27–7.06 | 0.012§ |
| *No. of items of key verbal discharge information*** | | | | *No. of items of key verbal discharge information*** | | | |
| 0 to 2 items | 3.18 | 0.15–67.18 | 0.458 | 0 to 2 items | 1.60 | 0.37–6.89 | 0.525 |
| **Hospital readmission within 5 weeks of discharge** | | | | **Hospital readmission within 18 weeks of discharge** | | | |
| *No. of items of key documented discharge information†* | | | | *No. of items of key documented discharge information†* | | | |
| 0 to 2 items | 0.60 | 0.25–1.43 | 0.252 | 0 to 2 items | 0.75 | 0.42–1.34 | 0.324 |
| *No. of items of key verbal discharge information*** | | | | *No. of items of key verbal discharge information*** | | | |
| 0 to 2 items | 0.73 | 0.30–1.75 | 0.479 | 0 to 2 items | 0.82 | 0.45–1.50 | 0.510 |
| **Self-reported deterioration of NCD/s within 5 weeks of discharge** | | | | **Self-reported deterioration of NCD/s within 18 weeks of discharge** | | | |
| *No. of items of key documented discharge information†* | | | | *No. of items of key documented discharge information†* | | | |
| 0 to 2 items | 1.41 | 0.70–2.86 | 0.338 | 0 to 2 items | 1.59 | 0.89–2.85 | 0.115 |
| *No. of items of key verbal discharge information*** | | | | *No. of items of key verbal discharge information*** | | | |
| 0 to 2 items | 0.65 | 0.32–1.34 | 0.241 | 0 to 2 items | 0.46 | 0.25–0.83 | 0.010§ |

*Adjusted for the following independent variables: sex, age group (18-49/50-69/70yrs+), education level (up to primary school-level/secondary school-level/higher school-level or more), employment status (unemployed/employed/retired), usual time taken to reach hospital (<1 hour/1-4 hours/>4 hours), number of chronic NCDs (1/2/3/4) and hospital site (1/2/3).

† Odds ratios represent association with receipt of 0 to 2 items of key documented information on discharge notes

§ Statistically significant at p<0.05

** Odds ratios represent association with receipt of 0 to 2 items of key verbal information during discharge consultation

## Discussion

### Main findings

This study has described discharge communication for chronic NCD patients in three public hospitals in India. One of our main findings was that both verbal and documented communication between HCPs and patients was often limited. Regarding documented information, discharge notes were predominantly provided on forms within minimal structure and in some cases were hastily written on blank sheets of paper or prescription forms. Patient attitudes towards discharge documents were generally positive, with the majority feeling it was important to receive them to help with understanding/explaining their condition, claiming insurance and getting attended to faster when they see the next HCP. However, the contents of discharge notes varied notably, with only just under a third of patients received notes containing all items of key information (i.e. diagnosis, medication information, lifestyle advice and follow-up instructions).

In addition, whilst the majority of patients reported being told where to go for a follow-up HCP visit during discharge consultations, other verbal information appeared to vary notably between patients. In particular, only just over half of all patients recalled receiving information

about necessary ongoing treatment/management for their NCD/s and under a quarter recalled receiving lifestyle advice. A markedly small minority of patients (3%) recalled receiving all four items of key verbal information during discharge consultations (i.e. ongoing treatment/management information, medication information, lifestyle advice and follow-up instructions). In addition, just under a fifth of patients were assessed by researchers as having either a basic or little/no understanding of important discharge information and 13% of patients only planned to return to a HCP when they were unwell again. These findings indicate that a significant proportion of patients left the hospital with sub-optimal levels of discharge information and/or comprehension regarding their ongoing healthcare needs, which may have compromised their ability to adequately manage their chronic NCD/s. Given the role that patient-held documents play in facilitating handover communication between HCPs in the study areas of India, the deficiencies in documented information provision may have also affected continuity of patient care.

Overall, the results reflect the limited similar research from India that has evidenced unstructured and deficient HCP-patient communication at the point of discharge.[32, 35] They are also consistent with other LMIC-based studies that, via patient reports and record evaluations, have evidenced a lack of in-depth information provision during discharge and/or poor levels of patient understanding regarding post-discharge care requirements.[27–30] The provision of deficient documented discharge information may be of particular concern for patient self-management, as global literature (predominantly from high-income countries) has indicated that individuals can struggle to absorb the verbal information provided by HCPs during healthcare consultations.[56] Further, whilst there is a dearth of empirical research on HCP attitudes towards discharge in India, our research on outpatients and qualitative data regarding inpatients from the same study areas in India indicates that a paucity of time, available HCPs, training and guidelines are likely to be notable contributors to the suboptimal communication evidenced in this study.[37] The results may also be somewhat explained by a historical lack of communication training in medical education and a tendency for paternalistic physician behaviour in India.[57, 58] A dominant HCP communication style is likely to result in unmet information needs from patients, due to them feeling intimidated and unable to ask questions. Similar communicative issues have been identified across healthcare settings in other parts of Asia, despite an increasing desire from patients for more involvement.[59] A global review of discharge communication literature suggests that both patients and HCPs prefer practices that are relevant, concise and personalised.[60] However, in practice HCPs across numerous healthcare settings report not having enough time to perform comprehensive discharge consultations and instead prioritising inpatient medical care.[24, 61] Overall, the international evidence would suggest that the issues found in this study pose a serious challenge for the Indian public health system, given the importance of information exchange in ensuring the continuity and safety of healthcare.[62]

Furthermore, to the best of the authors' knowledge, this study is the first of its kind to investigate the association between quality of discharge communication and adverse health outcomes in India. In the adjusted analyses we found that chronic NCD patients who received low-quality discharge notes (i.e. containing 0–2 items of key information) were more likely to have died within five and eighteen weeks of follow-up compared to those who received higher-quality notes. Such findings reflect HIC research, which has repeatedly demonstrated a link between deficient discharge information transfer and an increased risk of adverse events.[4–7] We also found that patients who received low-quality verbal discharge communication were less likely to report an NCD-related deterioration in their health within eighteen weeks of follow-up compared to those who received higher-quality communication. This was an unexpected finding, which could be explained by the possibility that those provided with less

information at discharge were patients with less severe health issues (due to HCP time pressures/case prioritisation etc.) and were subsequently less likely to report a deterioration later on. Overall, it is not possible to say whether low-quality discharge communication caused some, or all, of the health outcomes observed in this study. The small scale of the research alongside the complex nature of factors affecting health outcomes means that the results should be interpreted cautiously and will require further validation. Nonetheless, given the importance of handover communication for continuity of safety of care, the imperative to improve the recording and transfer of key healthcare information remains.

## Strengths and limitations

A key strength of this study is the collection of data from a range of patients across multiple healthcare facilities in varying geographical settings, which has provided a representative sample of chronic NCD patients accessing public hospitals in two diverse states of India. This study has also provided key insight into the transfer of critical NCD patient information during public hospital discharge in India. Follow-up data can be challenging and resource-intensive to collect in LMIC settings for a number of reasons, including a lack of comprehensive/up-to-date directories to assist with locating addresses, difficulties in contacting individuals to confirm visit attendance (due to a lack of phone ownership/poor network coverage) and limited access to certain areas (due to challenging terrains and/or limited transportation infrastructure). Therefore, this study has also provided an invaluable opportunity to explore the experiences of NCD patients following hospital discharge in the community.

However, given the vastness of India and the complexity of healthcare systems across LMICs, generalisability to other settings may be done with caution. A limitation of this study is that the quality of discharge communication was predominantly assessed via patient recall, rather than direct observation. A lack of adequately recorded inclusion/exclusion rates for participation is also a limitation as this could not be reported. The involvement of six different researchers may have increased the likelihood of researcher bias and, as the same researchers completed all data collection, they were not blinded to the quality of discharge communication. However, each section of the questionnaire was immediately filed away after completion and not referred to again which, alongside the fact that each researcher collected large volumes of questionnaire data, reduced potential for further bias. Regarding limitations of the regression analyses, data regarding diagnostic accuracy was unavailable and, theoretically, associations may have arisen as a result of imprecision (alpha error), indirect causal links (e.g. the provision of information was a marker for other aspects of care) or a combination of these factors. The limited number of deaths means it is also unclear how well our models adjusted for confounding factors, so the findings must be interpreted with caution. In addition, our follow-up questionnaire did not capture further information on patients who had died, so it was not possible to know whether these patients had been readmitted. Therefore, we ran further sensitivity analyses based on the assumption that all patients who died had also been readmitted; whilst no significant associations were found, adjusted point estimates regarding associations with low-quality discharge notes all leaned in the direction of an increased likelihood of hospital readmission within five and eighteen weeks follow-up (see "S7 Appendix" for results and goodness-of-fit statistics).

## Next steps

To effectively address the issues surrounding ineffective communication between HCPs and patients during discharge consultations, there is a pressing need for structured HCP training and handover communication guidelines to be implemented across public healthcare

facilities–our outpatient handover research from the same study hospitals has indicated that these are currently missing.[63]

Extensive HIC-based research has shown that education, simulation-training, and communication tools are effective strategies for improving the quality of healthcare information transfer during transitions of care.[12] Communication tools can also be adapted to ensure they facilitate patient-centred care at the point of discharge, which takes into account patients' needs, desires and values and involves patients and carers in care/decision-making processes [12, 24] In addition, several LMIC studies have evidenced that introducing patient discharge educational materials and structured disease-specific discharge planning can improve HCP to patient communication, as well as patient satisfaction, patient/carer healthcare management knowledge and post-discharge health outcomes.[64–68] Given the predominantly paper-based systems in use across the study settings, well-structured and standardised HCP checklists, documents and patient-held record booklets are also likely to advance the quantity and quality of essential information transferred between HCPs and between HCPs and patients and have proven successful in HIC and LMIC settings.[35, 69–71] Co-creation of such materials with HCPs, patients, carers and other key stakeholders should be considered in order to enhance acceptability, function and utilisation.

Looking to the future, the introduction of electronic health information systems holds great promise for improving information exchange and overall quality of healthcare.[72] A review of computer-enabled discharge communication interventions has evidenced their impact in improving timeliness and accuracy of information as well as patient and HCP satisfaction across Europe and North America.[60] In India, the government has recently announced long-term plans to digitize health records and this is currently being set up in seven states, including Himachal Pradesh and Kerala.[73] In addition, Kerala is the first Indian state to undergo comprehensive e-health systems reforms across all public healthcare facilities.[74] However, these developments are presently in their early stages and will take some time to become integrated enough to effectively facilitate handover communication; patient-held medical documents remain in use as the predominant vehicle for information transfer throughout public healthcare. In addition, e-health developments will not address integrative challenges between public and private HCPs using different information systems. This may continue to compromise continuity of care for many patients visiting multiple HCPs. Further, technological advances will not necessarily address improvements regarding the quality of information exchanged between HCPs and patients.

## Conclusion

In conclusion, this study has found that the quality of discharge communication for chronic NCD patients visiting public healthcare facilities in Himachal Pradesh and Kerala states in India is currently suboptimal. As a consequence of this, many patients with ongoing health needs are leaving hospitals with insufficient levels of information and/or understanding to be able to facilitate continuity of care and adequately manage their NCD/s. The findings have also evidenced significant associations between low-quality discharge communication and health outcomes at five and eighteen weeks follow-up. Whilst these associations must be interpreted with caution, overall this study has highlighted a pressing need for the wide-scale implementation of structured protocols, documents and training to improve the exchange of key discharge information between HCPs and between HCPs and patients. With the rising burden of NCDs across India and other LMICs, the findings from this study are timely and crucial for effective health systems development. Regarding future research, additional in-depth investigation is required to elucidate the validity of relationships between discharge communication and

health outcomes. It is also important that further robust LMIC studies are conducted to continue exploring the critical factors affecting the continuity and safety of NCD patient care and develop sustainable, cost-effective interventions.

## Supporting information

**S1 Appendix. Copy of previous handover research study questionnaire used as basis for current study questionnaire.**
(PDF)

**S2 Appendix. Baseline characteristics and adverse health outcomes by study site.**
(PDF)

**S3 Appendix. Exemplar picture of a structured discharge slip.**
(PDF)

**S4 Appendix. Summary flowchart of participant inclusion and exclusion throughout the study.**
(PDF)

**S5 Appendix. Results of all unadjusted regression analyses.**
(PDF)

**S6 Appendix. Summary of goodness-of-fit test results for all adjusted multivariate analyses.**
(PDF)

**S7 Appendix. Summary of results and goodness-of-fit test results for all sensitivity analyses.**
(PDF)

## Acknowledgments

We would like to extend our thanks to all patients, healthcare staff and researchers who kindly took the time to participate in this project or collect data. We are also indebted and give thanks to the participating hospitals. Finally, we are very grateful to the Directors of Health from both Himachal Pradesh and Kerala States, India, for their assistance in facilitating this project. Without their support this research would not have been possible.

## Author Contributions

**Conceptualization:** Jeemon Panniyammakal, Prabhakaran Dorairaj, Paramjit Gill, Richard Lilford, Semira Manaseki-Holland.

**Data curation:** Claire Humphries, Suganthi Jaganathan, Paramjit Gill, Semira Manaseki-Holland.

**Formal analysis:** Claire Humphries, Malcolm Price.

**Funding acquisition:** Jeemon Panniyammakal, Sanjeev Singh, Prabhakaran Dorairaj, Paramjit Gill, Sheila Greenfield, Richard Lilford, Semira Manaseki-Holland.

**Investigation:** Suganthi Jaganathan, Semira Manaseki-Holland.

**Methodology:** Semira Manaseki-Holland.

**Project administration:** Suganthi Jaganathan, Jeemon Panniyammakal, Sanjeev Singh, Prabhakaran Dorairaj, Semira Manaseki-Holland.

**Supervision:** Jeemon Panniyammakal, Sanjeev Singh, Prabhakaran Dorairaj, Semira Manaseki-Holland.

**Visualization:** Claire Humphries.

**Writing – original draft:** Claire Humphries.

**Writing – review & editing:** Claire Humphries, Suganthi Jaganathan, Jeemon Panniyammakal, Sanjeev Singh, Prabhakaran Dorairaj, Malcolm Price, Paramjit Gill, Sheila Greenfield, Richard Lilford, Semira Manaseki-Holland.

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
