## [Decision Letter · Decision Letter 0]

11 Dec 2019

PONE-D-19-22862

INVESTIGATING CHRONIC DISEASE PATIENTS’ ATTITUDES AND EXPERIENCES OF DISCHARGE COMMUNICATION IN THREE HOSPITALS IN INDIA

PLOS ONE

Dear Sir,

Thank you for submitting your manuscript to PLOS ONE. After careful consideration, we feel that it has merit but does not fully meet PLOS ONE’s publication criteria as it currently stands. The paper deals with an interesting topic and we feel that it merits publication.  Therefore, we invite you to submit a revised version of the manuscript that addresses the points raised during the review process. Please adress the major concerns especially raised by the reviewer 2. 

We would appreciate receiving your revised manuscript by 13/01/202.  To enhance the reproducibility of your results, we recommend that if applicable you deposit your laboratory protocols in protocols.io, where a protocol can be assigned its own identifier (DOI) such that it can be cited independently in the future. For instructions see: http://journals.plos.org/plosone/s/submission-guidelines#loc-laboratory-protocols

We look forward to receiving your revised manuscript.

Kind regards,

Prof. Prabath  Nanayakkara, MD, PhD, FRCP

Academic Editor

PLOS ONE

Journal Requirements:

2. Please include additional information regarding the PILOT questionnaire used in the study and ensure that you have provided sufficient details that others could replicate the analyses. For instance, how many participants were included in this pilot testing and was the original questionnaire altered in any way following the trial?

Reviewers' comments:

Reviewer's Responses to Questions

**Comments to the Author**

1. Is the manuscript technically sound, and do the data support the conclusions?

Reviewer #1: Yes

Reviewer #2: Partly

Reviewer #3: Yes

2. Has the statistical analysis been performed appropriately and rigorously? 

Reviewer #1: Yes

Reviewer #2: I Don't Know

Reviewer #3: I Don't Know

3. Have the authors made all data underlying the findings in their manuscript fully available?

Reviewer #1: Yes

Reviewer #2: Yes

Reviewer #3: Yes

4. Is the manuscript presented in an intelligible fashion and written in standard English?

Reviewer #1: Yes

Reviewer #2: Yes

Reviewer #3: Yes

5. Review Comments to the Author

Reviewer #1: In general, I think this article is well-written and covers an important topic of quality of medical care. The authors investigated whether discharge information is appropriate provided and investigated associations between the handover quality and health outcomes.

Major comments:

I am not sure whether the discovered associations between the (experienced) quality of the handover and health outcomes (especially all cause mortality) are that relevant, but these are a big part of the manuscript.

As the authors also state in their discussion this relation has to be interpreted with caution as confounding bias (in my opinion) may play an important role. For instance, compliance to therapy has not been measured and I can only imagine the quality of handover being relevant in mortality if this is caused by an Adverse Event of the specific disease (instead of other causes).

I think it's up to the editor to decide if this needs a change in the manuscript.

Minor comments

- Regarding the methods oft his study I think it's a limitation not observing the verbal handover information. Now, the patients' recall is measured, which of course indicates how well patients understand the provided information; however, there might be some differences between what has been told and what is recalled by patients, which may have implications for quality improvement strategies. In addition, the teach-back method in my opinion could have been a better method instead of using multiple choice questions to measure recall.

- Concerning exlusion: are patients with cognitive impairment included as well (not stated they have been excluded)

- You may consider to describe the pilot study a bit more in detail: how many patients were involved? Or did only researchers optimize the questionnaire? In that case, I think the study could have been optimized with pilot testing/cognitive interviewing of patients

- Abbreviations do come up in tekst without previous being presented in (..). For example LMIC line 114, CHC/PHC/SHS line 171

- Can you add a reference to the questionnaire used as a basis?

Reviewer #2: SUMMARY

This study describes the discharge communication process in association with adverse health outcomes in India. It was a prospective study including non-communicable patients in three public hospitals. The study shows that discharge communication is often limited and the quality sub-optimal and associated with adverse health outcomes in 5 and 18 weeks after discharge.

OVERALL IMPRESSION

Overall this is an interesting and important topic that adds to existing knowledge, especially since it was conducted in a low and middle income country (India), where evidence is currently relatively scarce. However, there are some major and minor issues that the manuscript raised:

MAJOR ISSUES

Overall

o The title states: ‘patients’ attitudes and experiences of discharge communication..’, and this is also stated as the primary aim. Reading this, I was expecting a qualitative collection of this information, providing a comprehensive overview of patients’ positive and negative experiences with discharge communication, what their views and opinions regarding the concept are, etc. However, this aim only seems to be inventoried with one question in the questionnaire (from appendix 2: question 5) “Do you think it is important to get such a note at discharge? Why?”. For me, this question does not reflect the concepts ‘attitudes’ and ‘experiences’. With the other questions, in my opinion, the authors captured what patients remembered regarding the content of the discharge communication. Therefore, I would consider changing ‘attitudes’ and ‘experiences’ in the title and aims to ‘patient recall of verbal discharge communication’, as the authors already do in the results section. This would align the title and aim better with the actual outcomes, results, and conclusion.

o The primary aim is on the concept stated above ‘to investigate chronic NCD patient’ attitudes and experiences of discharge communication’. The secondary aim was ‘to explore the relationship between quality of discharge communication and health outcomes’. However, throughout the entire paper and especially in the discussion/conclusion section, the secondary aim receives way more attention, while the primary aim receives little to none, which raised some questions with me.

o Considering the changes in the past years with regards to healthcare policy (i.e. the use of electronic health information systems, like the authors state in the discussion), are these data and results (of dec 2014 – nov 2015) still relevant?

Introduction

o It could use some more explanation how the conclusion in the line 125-127 follows from the information in line 119-125.

o In the introduction there seems to be a lot of focus on communication between HCP’s and less on communication between HCP and the patient, while the latter is your study subject. In order to keep focus on the subject you are studying, I would focus more on the communication between HCP-patient, and/or explain the relation with between-HCP’s communication more.

Methods

o Were the patients recruited consecutively (24 hours a day?), what limitation was caused by the ‘available resources’?, line 195.

o In line 153-159 and figure 1. the authors provide an overview of the Indian public healthcare system, however I miss an explanation of how this relates and/or is of influence to the data and results. Potentially, this overview fits better in the discussion, if supplemented with the connection to this study.

o Line 164-185 gives a pretty comprehensive overview of the population, but this is also not discussed in the discussion: is the included sample representative for the population?

o What do the authors mean by ‘who had been admitted within 24 hour’ in line 197? Does that mean that patients were included within the first 24 hours of their admission?

o There is no explanation of whether patients with (known) cognitive decline or dementia were in- or excluded. This is important information, as the majority of included patients were aged 60 or older. Also, the study subject regards the patients’ recollection of the discharge communication, which of course could be highly influences by impaired cognition.

o There is no explanation of how illiterate patients filled out the questionnaire.

o In line 223-226 the authors state to have piloted the questionnaire: what was the conclusion of this pilot (was the questionnaire comprehensible and comprehensive, or did it need more adaption after the pilot?), and were the piloted cases included in the main study?

o Line 233: when looking at the datasheet in appendix 2; there seems to have been no data collected on patient experiences, but on patients’ recollection of information transfer.

o As line 245-250 addresses an important limitation of the study, this section should move to the discussion.

o Why are multivariable/multiple logistic regression performed? The authors mention one predictor (low quality of discharge communication) and not multiple, so I have some doubts whether this is the fitting statistical method.

o Unclear why the authors chose the Firth methods of logistic regression and whether this is the suitable method to use in this study. They state the reason was the small number of outcome measures, however, to my knowledge this is not a suitable reason to perform Firth method. The Firth method is used when the outcomes are very rare, and I am not sure whether these outcome measures are necessarily rare.

o Why was the predictor categorized dichotomously (low quality vs high quality), instead of 4 categories? Based on content of the categories I wonder whether all separate key-items should have the equal amount of weight. Does this mean that they all have the same amount of importance/impact? For example, shouldn’t appropriate medical information be more important than lifestyle advice?

Results

o Line 321-324 is about the understanding of patients, but how did the researchers test/judge whether patients/cares had a good understanding? This needs explanation in the methods section. Also, nowhere in the methods section is explained that carers could also be present an/or interviewed.

o The section ‘patient follow-up plans’ (line 350-357) was not elaborated on in the methods section, why was this measured and how does this add to answering the research questions?

o Line 408-411 (decreased odds of self-reported deterioration of NCD/s when patients’ had low quality verbal communication) : this shows an interesting, probably unexpected, finding: I miss a potential explanation for this.

o I am unsure whether the content of the sensitivity analysis really adds value.

o Line 459-462: the data does not support these claims, verbal leans in the other direction.

o Table 2: why were data of 120 patients missing regarding ‘patient/carer understanding of health condition post discharge’?

Discussion and conclusion

o Overall, I feel that many of the claims in the discussion are expressed too bold, which is not backed by the data and the results. For example: line 595-597: the authors have not shown significant associations with adverse health outcomes after 5 and 18 weeks; only death seemed to show a significant association with low quality documented discharge information.

o Line 527-541: I believe this section should move to the limitations as it is addressing a limitation of the study.

o I think the fact that there were 6 different researchers/coders might have caused some bias, so should be mentioned in the limitations.

o The next steps section is focusing on between HCP’s communication, I miss the link to between HCP-patient communication here.

MINOR

Introduction

o Line 109: add abbreviation LMICs, as the abbreviation is used in line 120.

o The South African study in line 116-118 is on discharge planning, not on communication. Explain why this is important information for this study.

Methods

o Line 171: abbreviations not explained in text

o Were interviews completed face-to-face or via telephone (line 231-231)?

Results

o Table 3: I believe the ‘*’ is at the wrong place?

o Line 389: supposed to be S6?

Discussion

o Line 523: compared to..

o Line 546-548: why? explain

Reviewer #3: Summary of the research

The aim of the study was to analyze patients’ experiences of discharge communication after presenting with a chronic disease in one of the three participating Indian hospitals. The second aim was to find out whether the quality of discharge communication could be correlated to certain adverse events.

Overall impression

- The paper is well written. The background literature and study rationale are clearly articulated. Conclusions are consistent with the evidence and arguments presented. The authors do address the main question posed.

- The collected data exceeds the minimal sample size needed as calculated by the authors.

- This is a multicenter study including three public hospitals in India (one rural secondary-care hospital, one peri-urban secondary-care hospital and one Urban tertiary-care hospital)

- Research ethics (e.g. participant consent, ethics approval) are addressed appropriately.

- The overall methods are clear, although I’m missing some details to be able to replicate the study (see: minor issues)

- Figures and tables are overall clearly presented

To improve the manuscript:

Minor issues

- P2 L40: low numbers should be spelled out > three instead of 3.

- P2 L43: Five instead of 5

- P2 L51 / L55: in general percentages should be preceded by whole numbers

- P8 L200: Do the authors have any information concerning the amount of potentially eligible patients and excluded patients? Where patients included 7 days a week? During office hours?

- P9 L227: Were patients interviewed or did they fill out the questionnaires themselves? Was this done in the same way for every time point?

- P9 L229: Why did the authors choose to interview patients at 5 and 18 weeks after discharge?

- P12 L288: Add percentage after whole numbers.

- P13 Table 1: ‘Time taken to reach hospital >1 hour’ should be < 1hour

- P13 Table 1: ‘Death’ meaning all-cause mortality? Of NCD specific?

- S4 Appendix: Almost all died patients were included at site 3, although this site only included half the amount of patients in comparison to site 1. Do the authors of the paper have an explanation for this?

6. PLOS authors have the option to publish the peer review history of their article (what does this mean?). If published, this will include your full peer review and any attached files.

Reviewer #1: No

Reviewer #2: No

Reviewer #3: No

---

## [Author Response · Author response to Decision Letter 0]

25 Feb 2020

Manuscript ID: PONE-D-19-22862

Manuscript Title: Investigating chronic disease patients’ attitudes and experiences of discharge communication in three hospitals in India

Dear Editors, 

Thank you very much for reviewing our revised manuscript. We thank the reviewers for their generous feedback on our submission and have revised the manuscript accordingly. 

Please find below a point-by-point response to reviewer’s comments - please note that the page and line numbers provided in each response correspond to the unmarked main document. We hope that you find our responses satisfactory and that the manuscript is now acceptable for publication. 

With kind regards,

Dr. Semira Manaseki-Holland & Ms. Claire Humphries (BSc, MPH)

Institute of Applied Health Research

College of Medical and Dental Sciences

University of Birmingham

On behalf of all co-authors: 

R. Lilford, P. Dorairaj, P. Gill, J. Panniyammakal, S. Singh, S. Greenfield, M. Price, and S. Jaganathan.

Reviewer 1

Major comments:

1. I am not sure whether the discovered associations between the (experienced) quality of the handover and health outcomes (especially all cause mortality) are that relevant, but these are a big part of the manuscript. As the authors also state in their discussion this relation has to be interpreted with caution as confounding bias (in my opinion) may play an important role. For instance, compliance to therapy has not been measured and I can only imagine the quality of handover being relevant in mortality if this is caused by an Adverse Event of the specific disease (instead of other causes). I think it's up to the editor to decide if this needs a change in the manuscript.

Thank you for your comments. Whilst the authors acknowledge the limitations of the regression analyses within the manuscript, the findings are considered relevant as they are supported by a body of literature indicating that suboptimal discharge practices can compromise continuity and safety of patient care. 

Please note that amendments have been made to the discussion section of the manuscript to more suitably balance the coverage of results regarding the primary and secondary study objectives (see from line 475). 

Minor comments:

2. Regarding the methods of this study I think it's a limitation not observing the verbal handover information. Now, the patients' recall is measured, which of course indicates how well patients understand the provided information; however, there might be some differences between what has been told and what is recalled by patients, which may have implications for quality improvement strategies. In addition, the teach-back method in my opinion could have been a better method instead of using multiple choice questions to measure recall.

Thank you for your comments. The lack of observation of verbal handover information has now been noted within the limitations section (see lines 573-575). Regarding delivery of questionnaires, they were administered by researchers in the form of an interview, where each question was read aloud and researchers then ticked the appropriate box/es for responses that corresponded to pre-defined answers (rather than answers being read aloud). Researchers were also able to write free-text notes in designated spaces for responses that did not correspond to pre-defined answers. This information has now been clarified in the methods section of the manuscript (see from line 253). 

3. Concerning exclusion: are patients with cognitive impairment included as well (not stated they have been excluded)

Thank you for your query. Whilst there was no formal assessment of each patient’s cognitive abilities, researchers approached ward nurses to identify eligible patients. This process ensured that patients who were deemed too unwell to participate, due to severe physical and/or cognitive impairments, were excluded from the study. 

In addition, all patients came to hospital with a carer (i.e. friend, relative etc.). Therefore, whilst patients predominantly provided questionnaire responses, carers were available to provide support for answering questions for any patients who required it. 

The above information has now been clarified in the methods section of the manuscript (see from line 213). 

4. You may consider to describe the pilot study a bit more in detail: how many patients were involved? Or did only researchers optimize the questionnaire? In that case, I think the study could have been optimized with pilot testing/cognitive interviewing of patients

Thank you for your queries. Prior to the commencement of data collection, a small pilot study was conducted in Kerala to test all data collection instruments. This was an iterative process conducted over three rounds. Field workers went out to study hospitals with the questionnaires and interviewed two patients/carers each. As well as asking patients/carers questions from the questionnaires/topic guides, they also asked for feedback regarding the clarity and contextual appropriateness of the materials. Once this was complete, all researchers convened with a supervisor to discuss the intended aim of each question and patient/carer responses. During these discussions, the wording within the materials was developed to improve clarity and contextual relevance. This information has now been clarified within the methods section of the manuscript (see from line 235). 

5. Abbreviations do come up in text without previous being presented in (..). For example LMIC line 114, CHC/PHC/SHS line 171

Thank you for your comment- all abbreviations have now been presented in brackets prior to use throughout the manuscript. 

6. Can you add a reference to the questionnaire used as a basis?

Thank you for your query. The previous research involving the original questionnaire is unpublished and PLOS One does not allow unpublished work to be cited within an manuscript, so we are not able to reference it. Instead, we have included it in the supplementary material (see S2 Appendix) for your reference. 

Reviewer 2

Major comments: 

1. The title states: ‘patients’ attitudes and experiences of discharge communication..’, and this is also stated as the primary aim. Reading this, I was expecting a qualitative collection of this information, providing a comprehensive overview of patients’ positive and negative experiences with discharge communication, what their views and opinions regarding the concept are, etc. However, this aim only seems to be inventoried with one question in the questionnaire (from appendix 2: question 5) “Do you think it is important to get such a note at discharge? Why?”. For me, this question does not reflect the concepts ‘attitudes’ and ‘experiences’. With the other questions, in my opinion, the authors captured what patients remembered regarding the content of the discharge communication. Therefore, I would consider changing ‘attitudes’ and ‘experiences’ in the title and aims to ‘patient recall of verbal discharge communication’, as the authors already do in the results section. This would align the title and aim better with the actual outcomes, results, and conclusion.

Thank you for your comments. The authors acknowledge that there is a predominant focus on recall and direct assessment of information transfer at discharge rather than an exploration of patients’ attitudes and experiences of discharge communication. This has now been more accurately reflected in the study title, as well as the aims and objectives. 

2. The primary aim is on the concept stated above ‘to investigate chronic NCD patient’ attitudes and experiences of discharge communication’. The secondary aim was ‘to explore the relationship between quality of discharge communication and health outcomes’. However, throughout the entire paper and especially in the discussion/conclusion section, the secondary aim receives way more attention, while the primary aim receives little to none, which raised some questions with me.

Thank you for your comments. The authors disagree that the primary aim received little to no attention in the original manuscript; for example, lines 468-518 within the discussion were dedicated to findings regarding the primary objective and interpreting them in the context of the wider available literature. The sub-optimal nature of discharge communication and its consequences were also reported within the conclusion of the discussion section of the manuscript (lines 593-596). However, it is acknowledged that the secondary objective received a notable amount of attention and so amendments have been made to the discussion section of the manuscript to more suitably balance the coverage of findings in relation to the primary and secondary study objectives (see from line 477). 

3. Considering the changes in the past years with regards to healthcare policy (i.e. the use of electronic health information systems, like the authors state in the discussion), are these data and results (of dec 2014 – nov 2015) still relevant?

Thank you for your query. Although healthcare policy regarding electronic health information systems has passed, only the state of Kerala has started to implement it. At present there is little central government money and most state governments are expected to fund their own systems implementation, which is a situation that is unlikely to be practical for most states in the near future (including Himachal Pradesh). In addition, within Kerala the electronic systems that are in place are currently used as more of a method for recording service utilisation and prescribing. They remain in their early stages and are far from the full integration that could effectively facilitate handover communication between HCPs. Therefore, the practices described in the study areas remain widely the same, with patient-held notes still in use throughout public healthcare and particularly at the point of hospital discharge. As a result, the authors remain firmly of the belief that the data and results described within the manuscript are still relevant. 

Some further information clarifying current practices and progress of healthcare policy developments has been provided in the next steps of the discussion section of the manuscript (see from line 621). 

4. It could use some more explanation how the conclusion in the line 125-127 follows from the information in line 119-125.

Thank you for your comment. Further information explaining how patients facilitate information transfer between healthcare providers within the study areas of India has now been provided in the introduction section of the manuscript (see from line 129). 

5. In the introduction there seems to be a lot of focus on communication between HCP’s and less on communication between HCP and the patient, while the latter is your study subject. In order to keep focus on the subject you are studying, I would focus more on the communication between HCP-patient, and/or explain the relation with between-HCP’s communication more.

Thank you for your comments. The introduction section of the manuscript has been amended to more clearly demonstrate the significance and evidence regarding communication between HCPs and between HCPs and patients at the point of hospital discharge. In addition, further explanation has been provided as to why the study focussed on communication between HCPs and patients (see from line 90). 

6. Were the patients recruited consecutively (24 hours a day?), what limitation was caused by the ‘available resources’?, line 195.

Thank you for your queries. Patients were recruited consecutively by trained social work graduate researchers six days per week (spread across all days of the week over the study period) between the hours of 8am and 6pm, as this is the window within which patients were typically discharged from study hospitals. 

The wording “based on available resources” was simply used to demonstrate that as many patients were recruited as possible given the available human resources (i.e. six researchers). 

Information regarding patient recruitment has now been clarified in the methods section of the manuscript (see from line 207). 

7. In line 153-159 and figure 1. the authors provide an overview of the Indian public healthcare system, however I miss an explanation of how this relates and/or is of influence to the data and results. Potentially, this overview fits better in the discussion, if supplemented with the connection to this study.

Thank you for your comments. This was included within the main manuscript to provide general contextual information regarding the structure and functioning of public healthcare in India for readers who may not be familiar. It also gives an idea of the recommended coverage of each level of healthcare facility, which links to the next section regarding study populations that describes actual public primary care infrastructure in each area. The authors feel that this overview is better suited in the methods section, where the full text and figure can either be included in the manuscript or referred to via an appendix (depending on reviewer and/or editorial preference). 

8. Line 164-185 gives a pretty comprehensive overview of the population, but this is also not discussed in the discussion: is the included sample representative for the population?

Thank you for your query. This information was again provided for contextual purposes and can either remain in full within the methods section or be referred to via an appendix (depending on reviewer and/or editorial preference). Given a lack of available data regarding the prevalence of chronic NCD patients and their demographic details in the study areas of India at the time of the study, it was not possible to make exact calculations to guarantee a fully representative sample. However, recruiting as many individuals as possible from healthcare facilities over a period of several months helped to ensure that the study sample was representative of chronic NCD patients visiting public hospitals within Himachal Pradesh and Kerala states, India. This was demonstrated by the recruitment of patients with a range of chronic NCDs, ages, education levels, employment statuses etc. 

Some further information has been provided regarding representativeness of the study sample in the strengths of the discussion section of the manuscript (see lines 559-562). 

9. What do the authors mean by ‘who had been admitted within 24 hour’ in line 197? Does that mean that patients were included within the first 24 hours of their admission?

Thank you for your query. Patients were included within the first 24 hours of their admission. This information has now been clarified in the methods section of the manuscript (see lines 210-211). 

10. There is no explanation of whether patients with (known) cognitive decline or dementia were in- or excluded. This is important information, as the majority of included patients were aged 60 or older. Also, the study subject regards the patients’ recollection of the discharge communication, which of course could be highly influences by impaired cognition.

Thank you for your comments. Whilst there was no formal assessment of each patient’s cognitive abilities, researchers approached ward nurses to identify eligible patients. This process ensured that patients who were deemed too unwell to participate, due to severe physical and/or cognitive impairments, were excluded from the study. In addition, all patients came to hospital with a carer (i.e. friend, relative etc.). Therefore, whilst patients predominantly provided questionnaire responses, available carers could provide support for answering questions for those patients who required it. This information has now been clarified in the methods section of the manuscript (see from line 213). 

11. There is no explanation of how illiterate patients filled out the questionnaire.

Thank you for your comment. Questionnaires were administered in the form of an interview and responses were completed by researchers, which meant that illiterate patients were not excluded from participation. This has now been clarified in the methods section of the manuscript (see from line 253). 

12. In line 223-226 the authors state to have piloted the questionnaire: what was the conclusion of this pilot (was the questionnaire comprehensible and comprehensive, or did it need more adaption after the pilot?), and were the piloted cases included in the main study?

Thank you for your queries. Prior to the commencement of data collection, a small pilot study was conducted in Kerala to test all data collection instruments. This was an iterative process conducted over three rounds. Field workers went out to study hospitals with the questionnaires and interviewed two patients/carers each. As well as asking patients/carers questions from the questionnaires/topic guides, they also asked for feedback regarding the clarity and contextual appropriateness of the materials. Once this was complete, all researchers convened with a supervisor to discuss the intended aim of each question and patient/carer responses. During these discussions, the wording and structure of the questionnaire was developed to improve clarity and contextual relevance. After the three rounds of minor amendments, based on patient/carer and researcher feedback, the questionnaire was considered to be comprehensible and comprehensive. The piloted cases were not included in the main study. This information has now been clarified within the methods section of the manuscript (see from line 239). 

13. Line 233: when looking at the datasheet in appendix 2; there seems to have been no data collected on patient experiences, but on patients’ recollection of information transfer.

Thank you for your comment. The authors acknowledge reviewer comments regarding the focus on information transfer rather than patients’ attitudes and experiences of discharge communication. This has now been more accurately reflected in the study title as well as the aims and objectives. 

The authors also feel it is important to note that whilst the majority of data was based on patient recall, the data regarding documented information was extracted directly from medical notes (with patient permission) by researchers. 

14. As line 245-250 addresses an important limitation of the study, this section should move to the discussion.

Thank you for your comment. This section has now been moved into the discussion section of the manuscript (see from line 577). 

15. Why are multivariable/multiple logistic regression performed? The authors mention one predictor (low quality of discharge communication) and not multiple, so I have some doubts whether this is the fitting statistical method.

Thank you for your query. Multiple logistic regression was performed as the authors wished to try to adjust for potentially confounding variables. In the original manuscript, we stated from line 277: ‘Multivariable models adjusted for the following potential confounders: sex, age, education level, employment status, time taken to reach hospital, number of chronic NCDs and hospital site.’ Some further clarification has now been provided in the methods section of the manuscript (see lines 306-308). 

16. Unclear why the authors chose the Firth methods of logistic regression and whether this is the suitable method to use in this study. They state the reason was the small number of outcome measures, however, to my knowledge this is not a suitable reason to perform Firth method. The Firth method is used when the outcomes are very rare, and I am not sure whether these outcome measures are necessarily rare.

Thank you for your comments. The authors disagree with the reviewer on this point. It is the (small) number of outcome events rather than the proportion (rarity) of the outcome that requires the Firth method.[1] In other words, the problem is not the rarity of the outcomes, but the possibility of a small number of cases on the rarer of the two outcomes.[2] The Firth method has become a relatively standard approach for analysing binary outcomes with smaller samples.[3] 

References: 

1. Firth D. Bias reduction of maximum likelihood estimates. Biometrika. 1993;80(1):27-38.

2. Allison P. Logistic regression for rare events. Statistical Horizons. 2012. Available from: https://statisticalhorizons.com/logistic-regression-for-rare-events

3. Puhr, R, Heinze, G, Nold, M, et al. Firth’s logistic regression with rare events: accurate effect estimates and predictions? Stat Med 2017; 36: 2302–231

17. Why was the predictor categorized dichotomously (low quality vs high quality), instead of 4 categories? Based on content of the categories I wonder whether all separate key-items should have the equal amount of weight. Does this mean that they all have the same amount of importance/impact? For example, shouldn’t appropriate medical information be more important than lifestyle advice?

Thank you for your queries. All four items of key information were selected based on common themes across relevant literature regarding critical details needed for facilitating post-discharge continuity and safety of care. The data were not included as a single four-category variable because multiple pieces of discharge information can be given at one time. Inclusion of each item of information as a separate dichotomous variable would be problematic. As well as leading to reduced power, the assumption of linearity on the logit scale would also be questionable. For example, the predicted change in log odds ratio for a patient due to receiving lifestyle advice, would be the same for a patient who received lifestyle advice only, as for a patient who received treatment/management information, medication information, lifestyle advice and follow-up instructions. This assumption would be untestable as there are insufficient data to include the six interaction terms that would be required to assess it. 

18. Line 321-324 is about the understanding of patients, but how did the researchers test/judge whether patients/cares had a good understanding? This needs explanation in the methods section. Also, nowhere in the methods section is explained that carers could also be present an/or interviewed.

Thank you for your query and comments. Researcher judgements were made by asking patients/carers to verbally explain what they understood of their condition and post-discharge care requirements and then checking this information against their discharge notes, or if this was not available/did not contain enough information, a ward nurse ward who was aware of their condition and care requirements. This information has now been clarified in the methods section of the manuscript and it has been explained that whilst patients predominantly provided responses to questions, available carers were able to provide support with answers when necessary (see from line 257). 

19. The section ‘patient follow-up plans’ (line 350-357) was not elaborated on in the methods section, why was this measured and how does this add to answering the research questions?

Thank you for your query. Patient attitudes and plans regarding follow-up care can be affected by healthcare provider advice and can ultimately affect continuity of care and patient outcomes. Therefore, information on patient-follow-up plans was collected to see how chronic disease patients planned to manage their ongoing care needs following the receipt of discharge consultation information. Ultimately, this information increased the authors’ understanding of whether patients were leaving hospital fully prepared to effectively manage their condition/s. 

20. Line 408-411 (decreased odds of self-reported deterioration of NCD/s when patients’ had low quality verbal communication) : this shows an interesting, probably unexpected, finding: I miss a potential explanation for this.

Thank you for your comment. In the original manuscript the following potential explanation was offered for this finding in the discussion (lines 533 – 536): “For verbal information, it might be that those who were provide with less information at discharge were patients with less severe health issues (due to HCP time pressures/case prioritisation etc.) and were therefore less likely to report a deterioration later on”. The discussion section of the manuscript now contains a clearer and succinct summary of potential explanations for the findings from the regression analyses (see from line 536). 

21. I am unsure whether the content of the sensitivity analysis really adds value.

Thank you for your comment. The authors take the view that the sensitivity analyses adds some value by addressing a limitation of the study. In order to make the manuscript more concise, more detailed information regarding the sensitivity analyses and results, which was previously part of the results section of the manuscript, has now been moved into an appendix (see supplementary material “S9 Appendix); This is referred to in the limitations of the discussion section of the manuscript (see from line 591). 

22. Line 459-462: the data does not support these claims, verbal leans in the other direction.

Thank you for your comment. The sentence in question was referring to documented discharge information only and stated the following: “While no significant associations were found, all point estimates leaned in the direction of an increased likelihood of hospital readmission within five and eight weeks for those patients who received low-quality discharge notes”. The section of text regarding the sensitivity analyses has now been moved to an appendix and the description of results has been re-worded to improve clarity (see supplementary material “S9 Appendix”). 

23. Table 2: why were data of 120 patients missing regarding ‘patient/carer understanding of health condition post discharge’?

Thank you for your query. A footnote below table 2 explains the following: “please note the larger number of missing responses to this question was due to a lack of available documented discharge information and/or ward nurses, which were required at the time of questioning for the researcher to verify patient/carer understanding of their condition and post-discharge care requirements”. Further information regarding how patient/carer understanding data was collected has now been clarified in the methods section of the manuscript (see from line 267). 

24. Overall, I feel that many of the claims in the discussion are expressed too bold, which is not backed by the data and the results. For example: line 595-597: the authors have not shown significant associations with adverse health outcomes after 5 and 18 weeks; only death seemed to show a significant association with low quality documented discharge information.

Thank you for your comments. The discussion section of the manuscript has been amended to ensure the wording accurately represents the results and limitations of the study and it is maintained that the results of the regression analyses must be interpreted with caution (see from line 536).

25. Line 527-541: I believe this section should move to the limitations as it is addressing a limitation of the study.

Thank you for your comment. Further information regarding the limitations of the regression analyses has now been included in limitations of the discussion section of the manuscript (see from line 583). 

26. I think the fact that there were 6 different researchers/coders might have caused some bias, so should be mentioned in the limitations.

Thank you for your comment. The possibility of researcher bias has now been listed as a potential limitation of the study in the discussion section of the manuscript (see lines 577-578). 

27. The next steps section is focusing on between HCP’s communication, I miss the link to between HCP-patient communication here.

Thank you for your comment. Whilst the next steps section in the original manuscript did cover communication between HCPs and patients (e.g. in lines 567-570), the authors acknowledge that this could be made clearer. Therefore, to avoid confusion, the next steps within the discussion section of the manuscript has been amended to more clearly demonstrate suggestions for improving communication between HCPs and patients (see from line 598). 

Minor comments: 

28. Line 109: add abbreviation LMICs, as the abbreviation is used in line 120.

Thank you for your comment. The abbreviation has now been added (see line 112). 

29. The South African study in line 116-118 is on discharge planning, not on communication. Explain why this is important information for this study.

Thank you for your comment. The South African study was described to demonstrate previous LMIC findings regarding the significance of discharge processes for patient outcomes. It was considered relevant as discharge planning is a process that involves the transfer of healthcare information between HCPs and between HCPs and patients to ensure coordination and continuity of care; In the literature it has been defined as: “An interdisciplinary approach to continuity of care; it is a process that includes identification, assessment, goal setting, planning, implementation, coordination, and evaluation and is the quality link between hospitals, community-based services, nongovernment organizations, and carers”[1]. 

Please note that the introduction section of the manuscript has been amended to improve relevance and clarity (see from line 81). 

Reference:

1. Lin CJ, Cheng SJ, Shih SC, Chu CH, Tjung JJ. Discharge planning. International Journal of Gerontology. 2012 Dec 1;6(4):237-40.

30. Line 171: abbreviations not explained in text

Thank you for your comment. The abbreviations have now been explained in the text (see from line 182). 

31. Were interviews completed face-to-face or via telephone (line 231-231)?

Thank you for your query. All interviews were completed in person (i.e. face-to-face). This has now been clarified in the methods section of the manuscript (see from line 253). 

32. Table 3: I believe the ‘*’ is at the wrong place?

Thank you for your query. All symbols and corresponding footnotes for table 3 have now been checked for accuracy and corrected where necessary. 

33. Line 389: supposed to be S6?

Thank you for your query. All references to supplementary materials throughout the manuscript have been checked and corrected where necessary.

34. Line 523: compared to.

Thank you for comment. This line has now been clarified within the discussion section of the manuscript (see from line 541). 

35. Line 546-548: why? Explain (r.e. why follow-up data can be challenging to collect in LMIC settings)

Thank you for your query. Follow-up community-based data can be particularly challenging to collect and subsequently resource-intensive in LMIC settings such as India for a number of reasons. Some of these include: 

• Difficulties in locating addresses due to a lack of comprehensive and up-to-date directories

• Inability to contact individuals to confirm attendance at follow-up visits due to a lack of phone ownership or limited network coverage

• Limited access to certain areas due to limited transportation infrastructure, challenging terrains and/or conflict/military presence.

This has now been clarified in the strengths and limitations in the discussion section of the manuscript (see from line 564). 

Reviewer 3

Minor comments: 

1. P2 L40: low numbers should be spelled out > three instead of 3.

Thank you for your comment. This has now been corrected in the abstract. 

2. P2 L43: Five instead of 5

Thank you for your comment. This has now been corrected in the abstract. 

3. P2 L51 / L55: in general percentages should be preceded by whole numbers

Thank you for your comment. This has now been corrected in the abstract. 

4. P8 L200: Do the authors have any information concerning the amount of potentially eligible patients and excluded patients? Were patients included 7 days a week? During office hours?

Thank you for your queries. Due to the dynamic nature of the study settings, it was not possible to record inclusion/exclusion rates for participation. This has been noted as a limitation in the limitations within the discussion section of the manuscript (see lines 575-577). 

Patient recruitment was completed by researchers 6 days per week (spread across all days of the week over the study period) between the hours of 8am and 6pm; these hours were selected as this is the period within which patients were typically discharged from study hospitals. Therefore, the authors suspect no bias in recruitment due to days of the week or hours of the day selected for recruitment. This information has now been clarified in the methods section of the manuscript (see from line 207). 

5. P9 L227: Were patients interviewed or did they fill out the questionnaires themselves? Was this done in the same way for every time point?

Thank you for your queries. Questionnaires were administered by researchers in the form of an interview, where each question was read aloud and researchers then ticked the appropriate box/es for responses that corresponded to pre-defined answers (rather than answers being read aloud). Researchers were also able to write free-text notes in designated spaces for responses that did not correspond to pre-defined answers. This was done in the same way for every time point. This information has now been clarified in the methods section of the manuscript (see from line 253).

6. P9 L229: Why did the authors choose to interview patients at 5 and 18 weeks after discharge?

Thank you for your query. The follow-up questionnaires captured information on patient health-seeking behaviour. Therefore, patients were interviewed at five and eighteen-week follow-up due to the need for chronic NCD patients with ongoing health needs to have revisited healthcare providers within a month or at least within three months following hospital discharge. 

7. P12 L288: Add percentage after whole numbers.

Thank you for your comment. All percentages within the text have the manuscript have now been rounded to whole numbers. 

8. P13 Table 1: ‘Time taken to reach hospital >1 hour’ should be < 1hour

Thank you for your comment. This correction has been made in table 1. 

9. P13 Table 1: ‘Death’ meaning all-cause mortality? Or NCD specific?

Thank you for your query. “Death” refers to all-cause mortality – this has now been clarified in table 1 and in the methods section of the manuscript (see lines 296-297). 

10. S4 Appendix: Almost all died patients were included at site 3, although this site only included half the amount of patients in comparison to site 1. Do the authors of the paper have an explanation for this?

Thank you for your query. Site 1 and site 2 were hospitals situated in relatively non-deprived urban and peri-urban areas of Kerala. In these areas, patients have good levels of health literacy, there is a variety of healthcare providers available and the terrain and infrastructure makes travel reasonably easy for patients. Conversely, site 3 was a hospital situated in a rural and hilly area of Himachal Pradesh state, where patients are more socioeconomically deprived, generally less health literate and have longer distances to travel to a more limited range of healthcare providers. In addition, the climate is much cooler in this area so winters can be particularly harsh, with accompanying colds and flu that may have affected participants more than in Kerala. However, as we have no available data of the causes or circumstances of the deaths recorded in the study, we did not place too much emphasis on them and did not include the speculations above in the discussion – the authors can do this if reviewers /editor would like this inserted.

---

## [Editor Report · Decision Letter 1]

2 Mar 2020

DISCHARGE COMMUNICATION FOR CHRONIC DISEASE PATIENTS IN THREE HOSPITALS IN INDIA

PONE-D-19-22862R1

Dear Dr. Manaseki-Holland.

We are pleased to inform you that your manuscript has been judged scientifically suitable for publication and will be formally accepted for publication once it complies with all outstanding technical requirements.

With kind regards,

Prof. Prabath Nanayakkara, MD, PhD, FRCP

Academic Editor

PLOS ONE
---

## [Editor Report · Acceptance letter]

26 Mar 2020

PONE-D-19-22862R1 

Investigating discharge communication for chronic disease patients in three hospitals in India 

Dear Dr. Manaseki-Holland:

I am pleased to inform you that your manuscript has been deemed suitable for publication in PLOS ONE. Congratulations! Your manuscript is now with our production department. 

With kind regards,

on behalf of

Dr. P W. B. Nanayakkara 

Academic Editor

PLOS ONE